# Plasma and Urine Metabolites Associated with Microperimetric Retinal Sensitivity in Age-Related Macular Degeneration

**DOI:** 10.3390/metabo15040232

**Published:** 2025-03-28

**Authors:** Krupa Sourirajan, Kevin Mendez, Ines Lains, Gregory Tsougranis, Haemin Kang, Georgiy Kozak, Augustine Bannerman, Roshni Bhat, Hanna Choi, Archana Nigalye, Ivana K. Kim, Demetrios G. Vavvas, David M. Wu, Liming Liang, John B. Miller, Joan W. Miller, Jessica Lasky-Su, Deeba Husain

**Affiliations:** 1Massachusetts Eye and Ear, Department of Ophthalmology, Harvard Medical School, Boston, MA 02114, USA; ksourirajan@meei.harvard.edu (K.S.); rekme@channing.harvard.edu (K.M.); ines_lains@meei.harvard.edu (I.L.); gtsougranis3994@gmail.com (G.T.); hkang13@meei.harvard.edu (H.K.); gkozak@meei.harvard.edu (G.K.); augustine_bannerman@meei.harvard.edu (A.B.); rbhat2@meei.harvard.edu (R.B.); hchoi20@meei.harvard.edu (H.C.); narchanak@gmail.com (A.N.); ivana_kim@meei.harvard.edu (I.K.K.); demetrios_vavvas@meei.harvard.edu (D.G.V.); david_wu@meei.harvard.edu (D.M.W.); john_miller@meei.harvard.edu (J.B.M.); joan_miller@meei.harvard.edu (J.W.M.); 2Channing Lab, Channing Division of Network Medicine, Department of Medicine, Brigham and Women’s Hospital, Harvard Medical School, Boston, MA 02114, USA; jessica.su@channing.harvard.edu; 3Systems Genetics and Genomics Unit, Channing Division of Network Medicine, Brigham and Women’s Hospital, Harvard Medical School, Boston, MA 02114, USA; lliang@hsph.harvard.edu

**Keywords:** age-related macular degeneration, AMD, microperimetry, metabolomics, plasma, urine

## Abstract

Background: Best-corrected visual acuity (BCVA) is the current gold standard of retinal function measurement but is not affected in early and intermediate forms of age-related macular degeneration (AMD). Increasing evidence suggests that microperimetry is a sensitive measure of visual function. This study sought to analyze the associations between plasma and urine metabolites and microperimetry in AMD. Methods: We included data on 363 eyes (95 controls, 268 AMD). Microperimetry was performed in patients with or without AMD using the Macular Integrity Assessment (MAIA) microperimetry system, employing a 37-point full-threshold protocol. Plasma and urine samples were analyzed via ultra-high-performance liquid chromatography–mass spectrometry. Multilevel mixed-effects linear models were used to assess associations between the metabolites and retinal sensitivity. Statistical significance was determined by considering the number of independent tests that accounted for 80% of the variance (ENT80). Results: We identified two plasma and seven urine metabolites, which were significantly associated with mean retinal sensitivity in AMD, and the key results include metabolites in the lysine metabolism pathway. Conclusions: To our knowledge, we present the first assessment of the associations between plasma and urinary metabolites and retinal microperimetry sensitivity in AMD. This work can reveal more insight into the pathogenesis of AMD.

## 1. Introduction

Age-related macular degeneration (AMD) is the leading cause of blindness in people over 50 years old. In the United States alone, around 20 million people have AMD, and nearly 200 million people have AMD worldwide [1]. Best-corrected visual acuity (BCVA) is currently the gold-standard measure to functionally assess AMD. However, BCVA is not affected until late in the disease process [2]. Recent research supports that contrast sensitivity, low luminance visual acuity (LLVA), cone-specific contrast, and microperimetry are more reliable measures of disease progression. In particular, microperimetry is a non-invasive diagnostic tool that assesses central retinal sensitivity by directing stimuli of light directly onto the retina while simultaneously tracking the fundus location using a scanning laser ophthalmoscope. Prior research has highlighted that retinal sensitivity, as assessed by microperimetry, is affected early in the disease, leading to the appropriateness of monitoring visual functional changes in AMD patients [3]. However, the mechanisms behind these changes remain only partially understood.

Our group’s prior work has demonstrated the potential of plasma and urine metabolomics to better understand AMD pathogenesis [4]. Metabolomics is the study of metabolites and molecules downstream of all the genetic transcription processes and is thought to closely mirror multifactorial diseases like AMD [5,6]. In particular, we have reported that patients with AMD have a distinct metabolomic profile compared to the controls, with differences across stages of the disease [7]. A urinary metabolomic profile of AMD has also been identified, even though this demonstrates weaker associations than those seen in plasma [8]. However, to our knowledge, the metabolomic profiles of microperimetric changes in AMD have not been assessed. This study aims to assess the metabolomic profiles associated with retinal sensitivity changes using microperimetry in AMD.

## 2. Materials and Methods

### 2.1. Study Design

This was a cross-sectional study derived from a prospectively designed longitudinal project performed at the Department of Ophthalmology of Massachusetts Eye and Ear (MEE), Harvard Medical School, Boston, United States [4,7]. It was conducted in accordance with the Health Insurance Portability and Accountability Act requirements and the tenets of the Declaration of Helsinki. The MEE institutional review board approved this study, and written informed consent was obtained from all participants in the study.

### 2.2. Inclusion and Exclusion Criteria

We recruited and consented patients with or without an AMD diagnosis at their appointments with their providers. Those who were confirmed with an AMD diagnosis were classified under the AMD group. Those who did not have an AMD diagnosis were classified as the control group. All subjects were above the age of 50 years. All subjects were excluded if presenting with any other vitreoretinal diseases, active uveitis, or ocular infection, a formal diagnosis of glaucoma with a cup-to-disc ratio greater than 0.8, a refractive error equal to or greater than 6 diopters of spherical equivalent, significant media opacities that precluded observation of the ocular fundus, a history of retinal surgery, a history of any ocular surgery or intraocular procedure (such as laser or intraocular injections) within the 90 days before enrollment, or a diagnosis of diabetes mellitus, with or without concomitant diabetic retinopathy.

### 2.3. Study Protocol

The study participants underwent a comprehensive eye examination. A standardized questionnaire was applied to all subjects, which included questions on demographics, current medications, and past medical history. After confirmed overnight fasting, for all study participants, blood samples were collected in the morning into a sodium-heparin tube and were centrifuged within 30 min (1500 rpm, 10 min, 20 °C), and plasma aliquots of 1.5 mL and 5 mL were transferred into sterile cryovials for storage at −80 °C. Similarly, urine samples were aliquoted and stored under the same conditions in 60 mL urine collection cups (ThermoFisher Scientific, Waltham, MA, USA). Plasma and urine samples were shipped to Metabolon, Inc.^®^ (Morrisville, NC, USA), maintaining frozen conditions throughout transit. All shipments were completed within 48 h, and the samples remained frozen until they were stored at −80 °C for processing. A schematic workflow of our study is provided in Appendix A.

### 2.4. AMD Grading

All AMD participants were imaged with non-stereoscopic, 7-field, color fundus photographs (CFP) (Topcon TRC-50DX; Topcon Corporation, Tokyo, Japan). The images were graded using the Age-Related Eye Disease Study (AREDS) 2 classification by two independent graders masked to the subject’s clinical data [4,7]. A senior clinician (DH) established the final categorization if there were disagreements.

### 2.5. MAIA Microperimetry Testing

As described in our previous work, MAIA microperimetry (CentreVue, Padova, Italy) equipment was used [9]. The microperimetry exam was conducted under dark conditions, and Goldmann III-sized stimuli were presented to the subject in different locations. The stimuli were presented against a four apostles (1.27 cd/m^2^) background, and the stimuli ranged from 0 to 36 dB. Through an automated eye-tracking system, the machine was able to correct for eye movement and guarantee that the retina location and light stimulus on the image corresponded during the entire examination. Before the exam began, all participants went through a suprathreshold 10–2 ‘learning’ microperimetry test. Shortly after, the expert test was selected, including 37 points (4-2 strategy), and had retinal sensitivities at the 1, 3-, and 5-degree radii while covering a 10-degree diameter circle. At each degree, there were 12 stimuli and 1 at the central fovea. During the test, a real-time fundus image was produced through the confocal scanning laser ophthalmoscope. The fixation changes were registered at a rate of 25 times per second throughout the examination. When the exam ended, the threshold retinal sensitivity and fixation index were both exported. The eye was excluded from the study if fixation was not centered properly in the fovea.

### 2.6. Metabolomic Profiling

At Metabolon, Inc^®^, the **plasma and urine samples** underwent a detailed nontargeted mass spectrometry analysis using ultra-high-performance liquid chromatography–tandem MS (UPLC-MS/MS) [10]. In brief, all methods utilized a Waters ACQUITY ultra-performance liquid chromatography (UPLC) and a Thermo Scientific Q-Exactive high resolution/accurate mass spectrometer, interfaced with a heated electrospray ionization (HESI-II) source and Orbitrap mass analyzer, which was operated at 35,000 mass resolution. The sample extract was dried and then reconstituted in solvents compatible with each of the four methods. Each reconstitution solvent contained a series of standards at fixed concentrations to ensure injection and chromatographic consistency. One aliquot was analyzed using acidic positive-ion conditions, chromatographically optimized for more hydrophilic compounds. In this method, the extract was gradient-eluted from a C18 column (Waters, Milford MA, USA, UPLC BEH C18-2.1 × 100 mm, 1.7 μm) using water and methanol containing 0.05% perfluoropentanoic acid (PFPA) and 0.1% formic acid (FA). Another aliquot was also analyzed using acidic positive-ion conditions; however, it was chromatographically optimized for more hydrophobic compounds. In this method, the extract was gradient-eluted from the same aforementioned C18 column using methanol, acetonitrile, water, 0.05% PFPA, and 0.01% FA and was operated at an overall higher organic content. Another aliquot was analyzed using basic negative-ion optimized conditions using a separate dedicated C18 column. The basic extracts were gradient-eluted from the column using methanol and water, however, with 6.5 mM Ammonium Bicarbonate at pH 8. The fourth aliquot was analyzed via negative ionization following elution from a HILIC column (Waters UPLC BEH Amide 2.1 × 150 mm, 1.7 μm) using a gradient consisting of water and acetonitrile with 10 mM Ammonium Formate, pH 10.8. The MS analysis alternated between MS and data-dependent MSnscans using dynamic exclusion. The scan range varied slightly between methods but covered 70–1000 *m*/*z*. **Bioinformatics:** The informatics system consisted of four major components: the Laboratory Information Management System (LIMS), the data extraction and peak identification software, data processing tools for QC and compound identification, and a collection of information interpretation and visualization tools for use by data analysts. The hardware and software foundations for these informatics components were the LAN backbone and a database server running the Oracle 10.2.0.1 Enterprise Edition. **LIMS:** The purpose of the Metabolon LIMS system was to enable fully auditable laboratory automation through a secure, easy-to-use, and highly specialized system. The scope of the Metabolon LIMS system encompasses sample accessioning, sample preparation, instrumental analysis and reporting, and advanced data analysis. All of the subsequent software systems are grounded in the LIMS data structures. This was modified to leverage and interface with the in-house information extraction and data visualization systems, as well as third-party instrumentation and data analysis software. **Data Extraction and Compound Identification:** Raw data were extracted, peak-identified, and QC processed using Metabolon’s hardware and software. These systems are built on a web service platform utilizing Microsoft’s NET technologies, which run on high-performance application servers and fiber-channel storage arrays in clusters to provide active failover and load-balancing. Compounds were identified by comparison to library entries of purified standards or recurrent unknown entities. **Curation:** A variety of curation procedures were carried out to ensure that a high-quality dataset was made available for statistical analysis and data interpretation. The QC and curation processes were designed to ensure accurate and consistent identification of true chemical entities and to remove those representing system artifacts, misassignments, and background noise. The Metabolon data analysts used proprietary visualization and interpretation software to confirm the consistency of peak identification among the various samples. Library matches for each compound were checked for each sample and corrected if necessary.

For **urine samples,** the osmolality measurement was conducted before the UPLC-MS/MS, using a small aliquot (~20 uL) of urine, which was monitored via freezing point suppression on a Fiske 210 Osmometer (Advanced Instruments) or an equivalent instrument, according to the manufacturer’s instructions. Osmolality was measured in a range from 0 to 2000 mOsm. If the samples were outside that range, they were diluted appropriately to be in that range before data acquisition. For samples with osmolality values between 200 and 1200, 50 µL of the sample was used for extraction. Urine samples were then analyzed with the same UPLC-MS/MS approach by Metabolon, as described above.

Since the samples were measured in batches, data normalization at Metabolon, Inc.^®^ involved adjusting the median of each compound within each batch of plasma and urine samples to one and then proportionally normalizing each data point [4]. All samples were then run through our standard quality control and data processing pipeline [4,7]. Metabolite peak areas were then log-transformed and Pareto-scaled, with the missing values imputed using a half-minimum approach by cohort. Principal component analysis (PCA) was subsequently performed to uncover any potential outliers. As illustrated in Appendix A, no significant outliers were detected in either the plasma or urine samples.

### 2.7. Statistical Analysis

Clinical data were described using standard approaches, such as the mean and standard deviation for continuous variables and percentages for the categorical variables.

Metabolomics data underwent our standard quality control procedures, including logarithmic transformation and Pareto scaling of metabolite peak areas, with missing values imputed using a half-minimum approach [6,11]. To investigate the associations between metabolite levels and retinal function, we applied multilevel mixed-effects linear models, utilizing the R package “lme4” [12]. This analytical approach was tailored to include microperimetry measurements from both eyes of each patient, along with their plasma and urine metabolite levels. The primary endpoint of our study was the average retinal sensitivity, a critical measure in understanding retinal function.

Analyses were performed for plasma and urine, both for all samples and then stratified by the controls and patients with AMD. The models were adjusted for age, gender, body mass index (BMI), smoking status, and AREDS formulation supplementation. Statistical significance was determined using the Effective Number of Tests (ENT), a method calculated by identifying the number of principal components required to explain 80% of the data variance (ENT80) [4,13,14], with *p*-values of 0.0013 (0.05/39) reported as statistically significant. The difference in average retinal sensitivity by AMD stage was based on the analysis of variance (ANOVA) test.

Additional details are available in the online supplement.

## 3. Results

### 3.1. Demographics

We included 363 eyes, 95 controls, and 268 with AMD (49 controls and 141 AMD patients) (Table 1). Among these, 151 eyes also had urine data: 42 controls and 109 with AMD (21 controls and 60 AMD patients) (Appendix A). Overall, the mean average sensitivity was 25.7 (SD = 5.26). The average age of our patients was 68.6 (SD = 7.58), and the average BMI was 27.4 (SD = 4.97). The cohort breakdown of ex-smokers to nonsmokers was nearly similar, 173 and 182, respectively, and 48.2% of our cohort reported that they did not take AREDS supplementation, while 51.8% said yes.

### 3.2. Average Sensitivity by Stage

MAIA microperimetry was performed for the cohort, resulting in a final measure of mean retinal sensitivity per test. Figure 1 presents the mean retinal sensitivity by stage in our cohort. The mean retinal sensitivity is shown to grow worse as the stage advances (*p* < 2.2 × 10^−16^). For the controls, the mean retinal sensitivity was 28.7 dB (SD = 1.73). The mean retinal sensitivity for early AMD was 27.1 dB (SD = 5.79). The mean retinal sensitivity for intermediate AMD was 26.1 dB (SD = 3.18. Finally, the mean retinal sensitivity for late AMD was 17.2 dB (SD = 8.05). No participants were excluded based on outlier detection using the 1.5 × IQR method.

### 3.3. Microperimetry and Plasma Metabolites

Table 2 presents the significant associations between plasma metabolites and mean retinal sensitivity based on ENT80 (all results are provided in Appendix A).

When looking at all included eyes, phosphate under the energy super pathway and oxidative phosphorylation sub-pathway demonstrated negative associations (β = −10.0, *p* = 3.72 × 10^−5^). Significant results were also seen for some amino acids, such as oxindolylalanine, in the tryptophan metabolism sub-pathway (β = 3.4, *p* = 0.0009). In a stratified analysis for the AMD group, significant associations were shown by phosphate (β = −11.2, *p* = 0.0005) and glutarylcarnitine (C5-DC) in the lysine metabolism sub-pathway (β = 3.0, *p* = 0.001). When assessing the controls, phosphate (β = −6.5, *p* = 0.0004) also demonstrated significant associations.

### 3.4. Microperimetry and Urine Metabolites

In the overall analysis of urine metabolites, detailed in Table 3, significant associations with a retinal sensitivity based on ENT80 were identified within several sub-pathways (all results are provided in Appendix A). Appendix A revealed all associations between urine metabolites and mean retinal sensitivity for all patients. Appendix A showed these respective associations for the controls, while Appendix A’s associations were stratified by AMD. Under lysine metabolism, both 2-oxoadipate (β = −0.7, *p* = 3.87 × 10^−5^) and N6-acetyllysine (β = −5.2, *p* = 0.0001) were significant, as well as 5-hydroxylysine (β = −1.8, *p* = 0.0001). In purine metabolism, urate showed a significant negative association (β = −6.1, *p* = 8.24 × 10^−5^). Ethylmalonate was significant within the leucine, isoleucine, and valine metabolisms (β = −2.9, *p* = 8.65 × 10^−5^), and 3-hydroxyanthranilate was significant in tryptophan metabolism (β = −2.0, *p* = 0.0008). Stratified by AMD, the significant metabolites in the sub-pathways mirrored some from the overall analysis: 2-oxoadipate in lysine metabolism (β = −0.7, *p* = 0.0002), ethylmalonate in the leucine, isoleucine, and valine metabolisms (β = −3.4, *p* = 0.0003), and urate in purine metabolism (β = −7.4, *p* = 0.0004) were all notable. Additionally, 5-hydroxylysine in lysine metabolism (β = −2.3, *p* = 0.0006), pregnenetriol disulfate in pregnenolone steroids (β = −2.3, *p* = 0.0009), and N6-acetyllysine in lysine metabolism (β = −5.7, *p* = 0.0009) were also significant in the AMD subgroup. No significant metabolites were identified in the control group.

## 4. Discussion

We present a cross-sectional analysis of the associations between the plasma and urine metabolomic profiles and microperimetric retinal sensitivity in a cohort of AMD patients and controls. Both plasma and urinary metabolites of the lysine pathway were associated with the microperimetric findings in the AMD cases, thus supporting that this pathway is likely important in the pathogenesis of the functional changes observed. On the other hand, plasma phosphate levels were shown to be associated with retinal sensitivity in both groups, thus suggesting that this metabolite may play a role in retinal sensitivity, both in health and disease states.

As shown in our results above, metabolites of the lysine pathway were associated with retinal sensitivity in patients with AMD. In plasma, in particular, associations were identified with glutarylcarnitine (C5-DC), an acylcarnitine [15]. This is in agreement with prior research that has reported elevated levels of acylcarnitines in AMD patients compared to the controls [16,17] and suggested these metabolites as possible AMD biomarkers. Acylcarnitines play an important role in transport in beta-oxidation and fatty acid oxidation [15,18,19]. Of note, fatty acid oxidation is important for the normal function of photoreceptors in the retina, thus likely explaining the associations in this work [20].

In urine samples, three additional metabolites of the lysine pathway (2-oxoadipate, 5-hydroxylysine, and N6-acetyllysine) were also significantly associated with average retinal sensitivity in AMD patients. This is in agreement with the work from both our group and a review article highlighting the role of this pathway in AMD [16,21]. Lysine was found in higher concentrations in patients with AMD when compared to the controls [16]. In particular, our group has reported that metabolites of lysine, isoleucine, and leucine are associated with the progression of AMD at three years [21]. Additionally, metabolites belonging to the lysine pathway have been identified in our group’s previous work with an association with AMD [4,7].

The identification of consistent associations with metabolites of the lysine pathway in both plasma and urine samples and AMD is important, as it further supports the relevance of this pathway in the pathogenesis of this disease. Also of note, the easy accessibility of urine provided important advantages for the future development of biomarkers [21]. Urine samples are non-invasive and easy to obtain, avoiding some of the discomfort associated with blood draw.

We also identified significant associations between phosphate and retinal sensitivity both in patients with AMD and the controls. Phosphate is a metabolite of the oxidative phosphorylation sub-pathway. Oxidative phosphorylation is a cellular process that reduces oxygen to produce ATP [22]. Oxidative stress is caused by elevated reactive oxygen species (ROS) levels that damage cellular regulation, and this occurs in the aging process [23,24]. Additionally, research has shown that aging and the sensitivity of the visual field are negatively associated [25]. Resulting oxidative damage can alter metabolic pathways, and prior research has demonstrated the crucial role that the resulting damage has on pathways such as those found in AMD [26]. In a further stratified analysis, we identified phosphate to be significant in the control patients’ plasma samples, as well, indicating that phosphate might be a driving force of microperimetry. Aging plays a crucial role in AMD pathogenesis [27], and although the beta coefficient of the phosphate metabolite is higher in AMD patients compared to the controls, the similarity of phosphate between AMD patients and the controls indicates that phosphate might be a metabolite indicative of standard of aging.

This current study has several limitations, specifically, a relatively small sample size. In particular, urine samples were only available for a subset of our cohort. It would have been ideal to conduct a sub-analysis of AMD patients by stage; however, our sample size limited this possibility. Despite this, to our knowledge, this is the first time an association of metabolomics with microperimetry in AMD patients has been demonstrated. Longitudinal assessments to assess disease progression will be important to validate the findings reported further. Another limitation is that we assessed mean retinal sensitivity. Individual sensitivity values and their specific correlation with anatomy findings could provide more specifics. Also of note, we did not assess renal function in this work, which may influence the reported results. Lastly, this work uses untargeted metabolomics, and further work is needed to validate these findings using targeted experiments.

## 5. Conclusions

To our knowledge, we report, for the first time, associations between plasma and urine metabolic profiles and average retinal sensitivity measured by microperimetry. In particular, our findings support that both plasma and urine lysine metabolites are associated with this functional measure in patients with AMD, supporting the role of this pathway in the disease’s pathogenesis. Future work targeting this pathway may thus contribute to mitigating the visual impairments associated with this common blinding disease.

## Figures and Tables

**Figure 1 metabolites-15-00232-f001:**
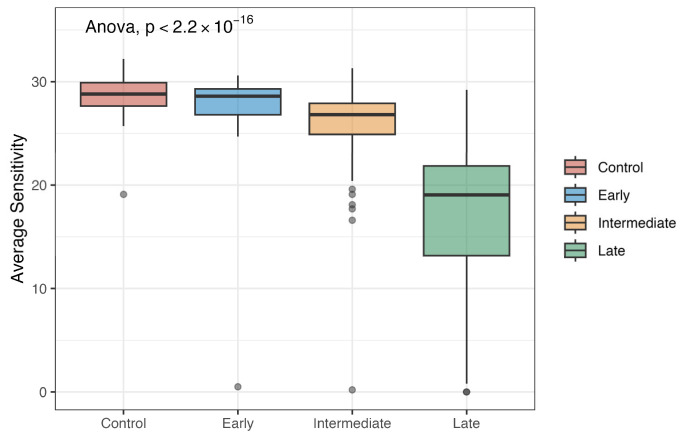
Boxplots of average sensitivity by AMD stage.

**Table 1 metabolites-15-00232-t001:** Demographic characterization for all study participants.

	0 (*N* = 95)	Early (*N* = 25)	Intermediate (*N* = 199)	Late (*N* = 44)	Overall (*N* = 363)
**Eye**					
** OD**	49 (51.6%)	15 (60.0%)	96 (48.2%)	24 (54.5%)	184 (50.7%)
** OS**	46 (48.4%)	10 (40.0%)	103 (51.8%)	20 (45.5%)	179 (49.3%)
**Average Sensitivity**					
** Mean (SD)**	28.7 (1.73)	27.1 (5.79)	26.1 (3.18)	17.2 (8.05)	25.7 (5.26)
** Median [Min, Max]**	28.8 [19.1,32.2]	28.6 [0.500, 30.6]	26.8 [0.200, 31.3]	19.1 [0, 29.2]	27.3 [0, 32.2]
**Age**					
** Mean (SD)**	64.7 (6.89)	66.7 (7.45)	69.9 (6.32)	72.6 (10.4)	68.6 (7.58)
** Median [Min, Max]**	65.0 [50.0, 83.0]	67.0 [54.0, 81.0]	70.0 [54.0, 86.0]	74.0 [45.0, 91.0]	69.0 [45.0, 91.0]
**BMI**					
** Mean (SD)**	26.8 (4.10)	27.2 (4.28)	27.7 (5.22)	27.2 (5.86)	27.4 (4.97)
** Median [Min, Max]**	25.8 [18.7, 42.9]	25.7 [22.0, 36.1]	26.5 [19.2, 53.0]	26.4 [19.2, 52.9]	26.3 [18.7, 53.0]
**Smoking**					
** Ex-smoker**	38 (40.0%)	7 (28.0%)	103 (51.8%)	25 (56.8%)	173 (47.7%)
** Nonsmoker**	55 (57.9%)	18 (72.0%)	90 (45.2%)	19 (43.2%)	182 (50.1%)
** Smoker**	2 (2.1%)	0 (0%)	6 (3.0%)	0 (0%)	8 (2.2%)
**AREDS**					
** No**	95 (100%)	22 (88.0%)	48 (24.1%)	10 (22.7%)	175 (48.2%)
** Yes**	0 (0%)	3 (12.0%)	151 (75.9%)	34 (77.3%)	188 (51.8%)

**Table 2 metabolites-15-00232-t002:** Plasma metabolites associated with mean retinal sensitivity based on ENT80 significance.

Analysis	Super Pathway	Sub-Pathway	Metabolite	Coefficient	*p*-Value
**All**	Energy	Oxidative Phosphorylation	Phosphate	−10.0	3.72 × 10^−5^
	Amino Acid	Tryptophan Metabolism	Oxindolylalanine	3.4	0.0009
**Controls**	Energy	Oxidative Phosphorylation	Phosphate	−6.5	0.0004
**AMD**	Energy	Oxidative Phosphorylation	Phosphate	−11.2	0.0005
	Amino Acid	Lysine Metabolism	Glutarylcarnitine (C5-DC)	3.0	0.001

AMD, age-related macular degeneration.

**Table 3 metabolites-15-00232-t003:** Urine metabolites associated with mean retinal sensitivity based on ENT80 significance.

Analysis	Super Pathway	Sub-Pathway	Metabolite	Coefficient	*p*-Value
**All**	Amino Acid	Lysine Metabolism	2-oxoadipate	−0.7	3.87 × 10^−5^
	Nucleotide	Purine Metabolism, (Hypo)Xanthine/Inosine containing	urate	−6.1	8.24 × 10^−5^
	Amino Acid	Leucine, Isoleucine, and Valine Metabolism	ethylmalonate	−2.9	8.65 × 10^−5^
	Amino Acid	Lysine Metabolism	N6-acetyllysine	−5.2	0.0001
	Amino Acid	Lysine Metabolism	5-hydroxylysine	−1.8	0.0001
	Amino Acid	Tryptophan Metabolism	3-hydroxyanthranilate	−2.0	0.0008
	Lipid	Progestin Steroids	5alpha-pregnan-3beta, 20alpha-diol disulfate	−1.4	0.001
	Carbohydrate	Aminosugar Metabolism	Erythronate *	−4.5	0.001
**AMD**	Amino Acid	Lysine Metabolism	2-oxoadipate	−0.7	0.0002
	Amino Acid	Leucine, Isoleucine, and Valine Metabolism	ethylmalonate	−3.4	0.0003
	Nucleotide	Purine Metabolism, (Hypo)Xanthine/Inosine containing	urate	−7.4	0.0004
	Amino Acid	Lysine Metabolism	5-hydroxylysine	−2.3	0.0006
	Lipid	Pregnenolone Steroids	pregnenetriol disulfate *	−2.3	0.0009
	Amino Acid	Lysine Metabolism	N6-acetyllysine	−5.7	0.001
	Carbohydrate	Aminosugar Metabolism	erythronate *	−5.8	0.001

AMD, age-related macular degeneration. * No results are reported for controls because no significant associations were observed in this subgroup.

## Data Availability

All tables and figures are original and have not been taken from any publication.

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
