# Peer review of "Plasma and Urine Metabolites Associated with Microperimetric Retinal Sensitivity in Age-Related Macular Degeneration"

_metabolites, 2025, doi:10.3390/metabo15040232_

Round 1
Reviewer 1 Report (Previous Reviewer 2)
Comments and Suggestions for Authors
The authors addressed my concerns properly
Reviewer 2 Report (Previous Reviewer 1)
Comments and Suggestions for Authors
All my questions have been answered, thank you.
This manuscript is a resubmission of an earlier submission. The following is a list of the peer review reports and author responses from that submission.
Round 1
Reviewer 1 Report
Comments and Suggestions for Authors
Very interesting work on microperimetry and metabolites in AMD which has been described clearly and concisely presented the results. I enjoyed reading the manuscript.
Yet I have a few remarks/questions:
line 124: You stated that some samples came from Coimbra in Portugal. Did you check for selection bias (samples from Portugal differs from those from MEE in terms of age/AMD stages of the patients or in the levels of the metabolites?) in addition to the batch bias that was adjusted (l.149)?
Did you conduct a subgroup analysis? It would be interesting to see if the plasma or urine level of the metabolites are different in low vs. advanced AMD.
Is there literature about taking AREDS formula and how it could affect metabolites in urine and blood samples? If so, could it be a counder that controls and patients with low AMD do not take AREDS formula at all in this study?
Author Response
Comment 1: “line 124: You stated that some samples came from Coimbra in Portugal. Did you check for selection bias (samples from Portugal differs from those from MEE in terms of age/AMD stages of the patients or in the levels of the metabolites?) in addition to the batch bias that was adjusted (l.149)?”
Response 1: Thank you for your question. While we did collect samples from Portugal, the Portugal subjects were not included in the analysis. We have updated the manuscript to reflect your comment [lines 128-129].
Comment 2: “Did you conduct a subgroup analysis? It would be interesting to see if the plasma or urine level of the metabolites are different in low vs. advanced AMD.”
Response 2: Thank you for your suggestion. We tried to conduct a subgroup analysis; however, given that the sample sizes were low, we were unable to get statistical significance when comparing early vs advanced AMD. We agree that it would be important to look at low vs advanced AMD if we had the greater sample size and we understand this is a limitation in our study. We have now addressed this in our manuscript [Lines 319-321].
Comment 3: “Is there literature about taking AREDS formula and how it could affect metabolites in urine and blood samples? If so, could it be a counder that controls and patients with low AMD do not take AREDS formula at all in this study?”
Response 3: Thank you for your comment. We agree that it would affect the metabolites in the urine and blood samples and that is why we adjusted for it in our models. AREDS supplementation contains various vitamins, including Vitamin E (tocopherol), which has been found in our previous work (PMID: 35050154). When people take vitamins, it can affect components of their metabolism (PMID: 31963141) . We have seen in our previous work that there is an “imbalance of AREDS formula supplement use by AMD stage (i.e., low supplementation in controls and early AMD, and high supplementation in intermediate and late AMD), combined with the small sample size”(PMID: 35268448). Based on our previous findings, we realized the importance of controlling for AREDS formula in our work.

Reviewer 2 Report
Comments and Suggestions for Authors
The manuscript first time presented a correlation study on plasma and urine metabolites and age-related macular degeneration (AMD). The correlation between the retinal sensitivity data acquired by microperimetry and the metabolomics data of urine and plasma was statistically analyzed by effective number of tests. The results suggested that metabolites of lysine pathway were associated in retinal sensitivity in patients with AMD. The data and statistical analysis results were properly reported, and the limitations of the work were properly addressed in the discussion sections. The quality of the work looks good to me. I don’t have more comments.
Author Response
Reviewer 2
Comment 1: “The manuscript first time presented a correlation study on plasma and urine metabolites and age-related macular degeneration (AMD). The correlation between the retinal sensitivity data acquired by microperimetry and the metabolomics data of urine and plasma was statistically analyzed by effective number of tests. The results suggested that metabolites of lysine pathway were associated in retinal sensitivity in patients with AMD. The data and statistical analysis results were properly reported, and the limitations of the work were properly addressed in the discussion sections. The quality of the work looks good to me. I don’t have more comments. “
Response 1: Thank you for taking the time to review this manuscript, we greatly appreciate it.
